# Statistical Topological Data Analysis – A Kernel Perspective

**Roland Kwitt**
Department of Computer Science
University of Salzburg
rkwitt@gmx.at

**Stefan Huber**
IST Austria
stefan.huber@ist.ac.at

**Marc Niethammer**
Department of Computer Science and BRIC
UNC Chapel Hill
mn@cs.unc.edu

**Weili Lin**
Department of Radiology and BRIC
UNC Chapel Hill
weili_lin@med.unc.edu

**Ulrich Bauer**
Department of Mathematics
Technische Universität München (TUM)
ulrich@bauer.org

## Abstract

We consider the problem of statistical computations with persistence diagrams, a summary representation of topological features in data. These diagrams encode persistent homology, a widely used invariant in topological data analysis. While several avenues towards a statistical treatment of the diagrams have been explored recently, we follow an alternative route that is motivated by the success of methods based on the embedding of probability measures into reproducing kernel Hilbert spaces. In fact, a positive definite kernel on persistence diagrams has recently been proposed, connecting persistent homology to popular kernel-based learning techniques such as support vector machines. However, important properties of that kernel enabling a principled use in the context of probability measure embeddings remain to be explored. Our contribution is to close this gap by proving universality of a variant of the original kernel, and to demonstrate its effective use in two-sample hypothesis testing on synthetic as well as real-world data.

## 1   Introduction

Over the past years, advances in adopting methods from algebraic topology to study the "shape" of data (e.g., point clouds, images, shapes) have given birth to the field of topological data analysis (TDA) [5]. In particular, persistent homology has been widely established as a tool for capturing "relevant" topological features at multiple scales. The output is a summary representation in the form of so called *barcodes* or *persistence diagrams*, which, roughly speaking, encode the life span of the features. These "topological summaries" have been successfully used in a variety of different fields, including, but not limited to, computer vision and medical imaging. Applications range from the analysis of cortical surface thickness [8] to the structure of brain networks [15], brain artery trees [2] or histology images for breast cancer analysis [22].

Despite the success of TDA in these areas, a statistical treatment of persistence diagrams (e.g., computing means or variances) turns out to be difficult, not least because of the unusual structure of the barcodes as intervals, rather than numerical quantities [1]. While substantial advancements in

the direction of statistical TDA have been made by studying the structure of the space of persistence diagrams endowed with $p$-Wasserstein metrics (or variants thereof) [18, 19, 28, 11], it is technically and computationally challenging to work in this space. In a machine learning context, we would rather work with Hilbert spaces, primarily due to the highly regular structure and the abundance of readily available and well-studied methods for statistics and learning.

One way to circumvent issues such as non-uniqueness of the Fréchet mean [18] or computationally intensive algorithmic strategies [28] is to consider mappings of persistence barcodes into linear function spaces. Statistical computations can then be performed based on probability theory on Banach spaces [14]. However, the methods proposed in [4] cannot guarantee that different probability distributions can always be distinguished by a statistical test.

**Contribution.** In this work, we consider the task of statistical computations with persistence diagrams. Our contribution is to approach this problem by leveraging the theory of embedding probability measures into reproducing kernel Hilbert spaces [23], in our case, probability measures on the space of persistence diagrams. In particular, we start with a recently introduced kernel on persistence diagrams by Reininghaus et al. [20] and identify missing properties that are essential for a well-founded use in the aforementioned framework. By enforcing mild restrictions on the underlying space, we can in fact close the remaining gaps and prove that a minor modification of the kernel is *universal* in the sense of Steinwart [25] (see Section 3). Our experiments demonstrate, on a couple of synthetic and real-world data samples, how this universal kernel enables a principled solution to the selected problem of (kernel-based) two-sample hypothesis testing.

**Related work.** In the following, we focus our attention on work related to a statistical treatment of persistent homology. Since this is a rather new field, several avenues are pursued in parallel. Mileyko et al. [18] study properties of the set of persistence diagrams when endowed with the $p$-Wasserstein metric. They show, for instance, that under this metric, the space is Polish and the Fréchet mean exists. However, it is not unique and no algorithmic solution is provided. Turner et al. [28] later show that the $L^2$-Wasserstein metric on the set of persistence diagrams yields a geodesic space, and that the additional structure can be leveraged to construct an algorithm for computing the Fréchet mean and to prove a law of large numbers. In [19], Munch et al. take a different approach and introduce a probabilistic variant of the Fréchet mean as a probability measure on persistence diagrams. While this yields a unique mean, the solution itself is not a persistence diagram anymore. Techniques for computing confidence sets for persistence diagrams are investigated by Fasy et al. [11]. The authors focus on the Bottleneck metric (i.e., a special case of the $p$-Wasserstein metric when $p = \infty$), remarking that similar results could potentially be obtained for the case of the $p$-Wasserstein metric under stronger assumptions on the underlying topological space.

While the aforementioned results concern properties of the set of persistence diagrams equipped with $p$-Wasserstein metrics, a different strategy is advocated by Bubenik in [4]. The key idea is to circumvent the peculiarities of the metric by mapping persistence diagrams into function spaces. One such representation is the persistence landscape, i.e., a sequence of 1-Lipschitz functions in a Banach space. While it is in general not possible to go back and forth between landscapes and persistence diagrams, the Banach space structure enables a well-founded theoretical treatment of statistical concepts, such as averages or confidence intervals [14]. Chazal et al. [6] establish additional convergence results and propose a bootstrap procedure for obtaining confidence sets.

Another, less statistically oriented, approach towards a convenient summary of persistence barcodes is followed by Adcock et al. [1]. The idea is to attach numerical quantities to persistence barcodes, which can then be used as input to any machine learning algorithm in the form of feature vectors. This strategy is rooted in a study of algebraic functions on barcodes. However, it does not necessarily guarantee stability of the persistence summary representation, which is typically a desired property of a feature map [20].

Our proposed approach to statistical TDA is also closely related to work in the field of kernel-based learning techniques [21] or, to be more specific, to the embedding of probability measures into a RKHS [23] and the study of suitable kernel functions in that context [7, 24]. In fact, the idea of mapping probability measures into a RKHS has led to many developments generalizing statistical concepts, such as two-sample testing [13], testing for conditional independence, or statistical inference [12], form Euclidean spaces to other domains equipped with a kernel. In the context of supervised learning with TDA, Reininghaus et al. [20] recently established a first connection to

kernel-based learning techniques via the definition of a positive definite kernel on persistence diagrams. While positive definiteness is sufficient for many techniques, such as support vector machines or kernel PCA, additional properties are required in the context of embedding probability measures.

**Organization.** Section 2 briefly reviews some background material and introduces some notation. In Section 3, we show how a slight modification of the kernel in [20] fits into the framework of embedding probability measures into a RKHS. Section 4 presents a set of experiments on synthetic and real data, highlighting the advantages of the kernel. Finally, Section 5 summarizes the main contributions and discusses future directions.

## 2 Background

Since our discussion of statistical TDA from a kernel perspective is largely decoupled from how the topological summaries are obtained, we only review two important notions for the theory of persistent homology: *filtrations* and *persistence diagrams*. For a thorough treatment of the topic, we refer the reader to [10]. We also briefly review the concept of embedding probability measures into a RKHS, following [23].

**Filtrations.** A standard approach to TDA assigns to some metric space $(\mathbb{M}, d_\mathbb{M})$ a growing sequence of simplicial complexes (indexed by a parameter $t \in \mathbb{R}$), typically referred to as a *filtration*. Recall that an abstract simplicial complex is a collection of nonempty sets that is closed under taking nonempty subsets. Persistent homology then studies the evolution of the homology of these complexes for a growing parameter $t$. Some widely used constructions, particularly for point cloud data, are the Vietoris–Rips and the Čech complex. The *Vietoris–Rips* complex is a simplicial complex with vertex set $\mathbb{M}$ such that $[x_0, \dots, x_m]$ is an $m$-simplex iff $\max_{i,j \leq m} d_\mathbb{M}(x_i, x_j) \leq t$. For a point set $\mathbb{M} \subset \mathbb{R}^d$ in Euclidean space, the *Čech complex* is a simplicial complex with vertex set $\mathbb{M} \subset \mathbb{R}^d$ such that $[x_0, \dots, x_m]$ is an $m$-simplex iff the closed balls of radius $t$ centered at the $x_i$ have a non-empty common intersection.

A more general way of obtaining a filtration is to consider the sublevel sets $f^{-1}(-\infty, t]$, for $t \in \mathbb{R}$, of a function $f : \mathbb{X} \to \mathbb{R}$ on a topological space $\mathbb{X}$. For instance, in the case of surfaces meshes, a commonly used function is the *heat kernel signature (HKS)* [27]. The Čech and Vietoris–Rips filtrations appear as special cases, both being sublevel set filtrations of an appropriate function on the subsets (abstract simplices) of the vertex set $\mathbb{M}$: for the Čech filtration, the function assigns to each subset the radius of its smallest enclosing sphere, while for the Vietoris–Rips filtration, the function assigns to each subset its diameter (equivalently, the length of its longest edge).

**Persistence diagrams.** Studying the evolution of the topology of a filtration allows us to capture interesting properties of the metric or function used to generate the filtration. Persistence diagrams provide a concise description of the changes in homology that occur during this process. Existing connected components may merge, cycles may appear, etc. This leads to the appearance and disappearance of homological features of different dimension. Persistent homology tracks the birth $b$ and death $d$ of such topological features. The multiset of points $p$, where each point $p = (b, d)$ corresponds to a birth/death time pair, is called the *persistence diagram* of the filtration. An example of a per-

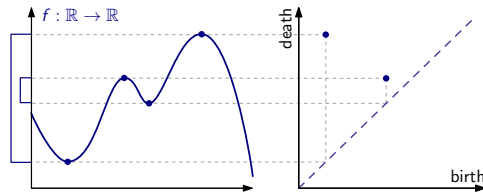

**Fig. 1:** A function and its 0-th persistence diagram.

sistence diagram for 0-dimensional features (i.e., connected components) of a function $f : \mathbb{X} \to \mathbb{R}$ with $\mathbb{X} = \mathbb{R}$ is shown in Fig. 2. We use the identifiers $F, G$ to denote persistence diagrams in the remainder of the paper. Since $d > b$, all points lie in the half-plane above the diagonal.

**RKHS embedding of probability measures**. An important concept for our work is the embedding of probability measures into reproducing kernel Hilbert spaces [23]. Consider a Borel probability measure $P$ defined on a compact metric space $(\mathcal{X}, d)$, which we observe through the i.i.d. sample $X = \{x_i\}_{i=1}^m$ with $x_i \sim P$. Furthermore, let $k : \mathcal{X} \times \mathcal{X} \to \mathbb{R}$ be a positive definite kernel, i.e., a function which realizes an inner product $k(x, y) = \langle \phi(x), \phi(y) \rangle_\mathcal{G}$ with $x, y \in \mathcal{X}$ in some Hilbert space $\mathcal{G}$ for some (possibly unknown) map $\phi : \mathcal{X} \to \mathcal{G}$ (see [26, Definition 4.1.]). Also, let $\mathcal{H}$ be the associated RKHS, generated by functions $k_x = k(x, \cdot) : \mathcal{X} \to \mathbb{R}$ induced by the kernel, i.e., $\mathcal{H} = \overline{\text{span}\{k_x : x \in \mathcal{X}\}} = \overline{\text{span}\{\langle \phi(x), \phi(\cdot) \rangle_\mathcal{G} : x \in \mathcal{X}\}}$, with the scalar product $\langle k_x, k_y \rangle_\mathcal{H} = k(x, y)$.

The linear structure on the RKHS $\mathcal{H}$ admits the construction of means. The embedding of a probability measure $P$ on $X$ is now accomplished via the *mean map* $\mu : P \mapsto \mu_P = \mathbf{E}_{x\sim P}[k_x]$. If this map is injective, the kernel $k$ is called *characteristic*. This is true, in particular, if $\mathcal{H}$ is dense in the space of continuous functions $\mathcal{X} \to \mathbb{R}$ (with the supremum norm), in which case we refer to the kernel as *universal* [25]. While a universal kernel is always characteristic, the converse is not true.

Since it has been shown [13] that the empirical estimate of the mean, $\mu_X = {}^1/m \sum_i k_{x_i}$, is a good proxy for $\mu_P$, the injectivity of $\mu$ can be used to define distances between distributions $P$ and $Q$, observed via samples $X = \{x_i\}_{i=1}^m$ and $Y = \{y_i\}_{i=1}^n$. Specifically, this can be done via the *maximum mean discrepancy*

$$\text{MMD}[\mathcal{F}, P, Q] = \sup_{f \in \mathcal{F}} (\mathbf{E}_{x\sim P}[f(x)] - \mathbf{E}_{y\sim Q}[f(y)]), \tag{1}$$

where $\mathcal{F}$ denotes a suitable class of functions $\mathcal{X} \to \mathbb{R}$, and $\mathbf{E}_{x\sim P}[f(x)]$ denotes the expectation of $f(x)$ w.r.t. $P$ (which can be written as $\langle \mu_P, f \rangle$ by virtue of the reproducing property of $k$). Gretton et al. [13] restrict $\mathcal{F}$ to functions on a unit ball in $\mathcal{H}$, i.e., $\mathcal{F} = \{f \in \mathcal{H} : \|f\|_\infty \leq 1\}$, and show that Eq. (1) can be expressed as the RHKS distance between the means $\mu_P$ and $\mu_Q$ of the measures $P$ and $Q$ as $\text{MMD}^2[\mathcal{F}, P, Q] = \|\mu_P - \mu_Q\|_{\mathcal{H}}^2$. Empirical estimates of this quantity are given in [13]. This connection is of particular importance to us, since it allows for two-sample hypothesis testing in a principled manner given a suitable (characteristic/universal) kernel. Prominent examples of universal kernels for $\mathcal{X} = \mathbb{R}^d$ are the Gaussian RBF kernel $k(x, y) = e^{-\gamma\|x-y\|^2}$ and the kernel $e^{\langle x, y \rangle}$. However, *without* a characteristic/universal kernel, $\text{MMD}[\mathcal{F}, P, Q] = 0$ does not imply $P = Q$. A well-known example of a non-characteristic kernel is the scalar product kernel $k(x, y) = \langle x, y \rangle$ with $x, y \in \mathbb{R}^d$. Even if $P \neq Q$, e.g., if the variances of the distributions differ, the MMD will still be zero if the means are equal.

In the context of a statistical treatment of persistent homology, the ability to embed probability measures on the space of persistence diagrams into a RKHS is appealing. Specifically, the problem of testing whether two different samples exhibit significantly different homological features – as captured in the persistence diagram – boils down to a two-sample test with null hypothesis $H_0 : \mu_P = \mu_Q$ vs. a general alternative $H_A : \mu_P \neq \mu_Q$, where $P$ and $Q$ are probability measures on the set of persistence diagrams. The computation of this test only involves evaluations of the kernel. Enabling this procedure via a suitable universal kernel will be discussed next.

## 3 The universal persistence scale space kernel

In the following, for $1 \leq q \leq \infty$ we let $\mathcal{D}_q = \{F \mid d_{W,q}(F, \emptyset) < \infty\}$, denote the metric space of persistence diagrams with the $q$-Wasserstein metric $d_{W,q}$[1], where $\emptyset$ is the empty diagram. In [18, Theorem 1], Mileyko et al. show that $(\mathcal{D}_q, d_{W,q})$ is a complete metric space. When the subscript $q$ is omitted, we do not refer to any specific instance of $q$-Wasserstein metric.

Let us fix the numbers $N \in \mathbb{N}$ and $R \in \mathbb{R}$. We denote by $\mathcal{S}$ the subset of $\mathcal{D}$ consisting of those persistence diagrams that are birth-death bounded by $R$ (i.e., for every $D \in \mathcal{S}$ the birth/death time of its points is less or equal to $R$; see [18, Definition 5]) and whose total multiplicities (i.e., the sum of multiplicities of all points in a diagram) are bounded by $N$. While this might appear restrictive at first sight, it does not really pose a limitation in practice. In fact, for data generated by some finite process (e.g., meshes have a finite number of vertices/faces, images have limited resolution, etc.), establishing $N$ and $R$ is typically not a problem. We remark that the aforementioned restriction is similar to enforcing boundedness of the support of persistence landscapes in [4, Section 3.6].

In [20], Reininghaus et al. introduce the *persistence scale space* (PSS) kernel as a stable, multi-scale kernel on the set $\mathcal{D}$ of persistence diagrams of finite total multiplicity, i.e., each diagram contains only finitely many points. Let $p = (b, d)$ denote a point in a diagram $F \in \mathcal{D}$, and let $\overline{p} = (d, b)$ denote its mirror image across the diagonal. Further, let $\Omega = \{x = (x_1, x_2) \in \mathbb{R}^2, x_2 \geq x_1\}$. The feature map $\Phi_\sigma : \mathcal{D} \to L^2(\Omega)$ is given as the solution of a heat diffusion problem with a Dirichlet boundary condition on the diagonal by

$$\Phi_\sigma(F) \colon \Omega \to \mathbb{R}, \quad x \mapsto \frac{1}{4\pi\sigma} \sum_{p \in F} e^{-\frac{\|x-p\|^2}{4\sigma}} - e^{-\frac{\|x-\overline{p}\|^2}{4\sigma}}. \tag{2}$$

The kernel $k_\sigma \colon \mathcal{D} \times \mathcal{D} \to \mathbb{R}$ is then given in closed form as

$$k_\sigma(F,G) = \langle \Phi_\sigma(F), \Phi_\sigma(G) \rangle_{L^2(\Omega)} = \frac{1}{8\pi\sigma} \sum_{\substack{p \in F \\ q \in G}} e^{-\frac{\|p-q\|^2}{8\sigma}} - e^{-\frac{\|p-\overline{q}\|^2}{8\sigma}}. \tag{3}$$

for $\sigma > 0$ and $F, G \in \mathcal{D}$. By construction, positive definiteness of $k_\sigma$ is guaranteed. The kernel is *stable* in the sense that the distance $d_\sigma(F,G) = \sqrt{k(F,F) + k(G,G) - 2k(F,G)}$ is bounded up to a constant by $d_{W,1}(F,G)$ [20, Theorem 2].

We have the following property:

**Proposition 1.** *Restricting the kernel in Eq. (3) to $\mathcal{S} \times \mathcal{S}$, the mean map $\mu$ sends a probability measure $P$ on $\mathcal{S}$ to an element $\mu_P \in \mathcal{H}$.*

*Proof.* The claim immediately follows from [13, Lemma 3] and [24, Proposition 2], since $k_\sigma$ is measurable and bounded on $\mathcal{S}$, and hence $\mu_P \in \mathcal{H}$. $\qquad\square$

While positive definiteness enables the use of $k_\sigma$ in many kernel-based learning techniques [21], we are interested in assessing whether it is universal, or if we can construct a universal kernel from $k_\sigma$ (see Section 2). The following theorem of Christmann and Steinwart [7] is particularly relevant to this question.

**Theorem 1.** *(cf. Theorem 2.2 of [7]) Let $\mathcal{X}$ be a compact metric space and $\mathcal{G}$ a separable Hilbert space such that there exists a continuous and injective map $\Phi \colon \mathcal{X} \to \mathcal{G}$. Furthermore, let $K \colon \mathbb{R} \to \mathbb{R}$ be a function that is analytic on some neighborhood of 0, i.e., it can locally be expressed by its Taylor series*

$$K(t) = \sum_{n=0}^\infty a_n t^n, \quad t \in [-r,r].$$

*If $a_n > 0$ for all $n \in \mathbb{N}_0$, then $k \colon \mathcal{X} \times \mathcal{X} \to \mathbb{R}$,*

$$k(x,y) = K(\langle \Phi(x), \Phi(y) \rangle_\mathcal{G}) = \sum_{n=0}^\infty a_n \langle \Phi(x), \Phi(y) \rangle_\mathcal{G}^n. \tag{4}$$

*is a universal kernel.*

Kernels of the form Eq. (4) are typically referred to as *Taylor kernels*.

Note that universality of a kernel on $\mathcal{X}$ refers to a specific choice of metric on $\mathcal{X}$. By using the same argument as for the linear dot-product kernel in $\mathbb{R}^d$ (see above), the PSS kernel $k_\sigma$ *cannot* be universal with respect to the metric $d_{k_\sigma}$, which is induced by the scalar product defining $k_\sigma$. On the other hand, it is unclear whether $k_\sigma$ is universal with respect to the metric $d_{W,q}$. However, we do have the following result:

**Proposition 2.** *The kernel $k_\sigma^U \colon \mathcal{S} \times \mathcal{S} \to \mathbb{R}$,*

$$k_\sigma^U(F,G) = \exp(k_\sigma(F,G)), \tag{5}$$

*is universal with respect to the metric $d_{W,1}$.*

*Proof.* We prove this proposition by means of Theorem 1. We set $\mathcal{G} = L^2(\Omega)$, which is a separable Hilbert space. As shown in Reininghaus et al. [20], the feature map $\Phi_\sigma \colon \mathcal{D} \to L^2(\Omega)$ is injective. Furthermore, it is continuous by construction, as the metric on $\mathcal{D}$ is induced by the norm on $L^2(\Omega)$, and so is $\Phi_\sigma$ restricted to $\mathcal{S}$. The function $K \colon \mathbb{R} \to \mathbb{R}$ is defined as $x \mapsto \exp(x)$, and hence is analytic on $\mathbb{R}$. Its Taylor coefficients $a_n$ are $1/n!$, and thus are positive for any $n$.

It remains to show that $(\mathcal{S}, d_{W,1})$ is a compact metric space. First, define $\mathcal{R} = \Omega^N \cap ([-R,R]^2)^N$, which is a bounded, closed, and therefore compact subspace of $(\mathbb{R}^2)^N$. Now consider the function

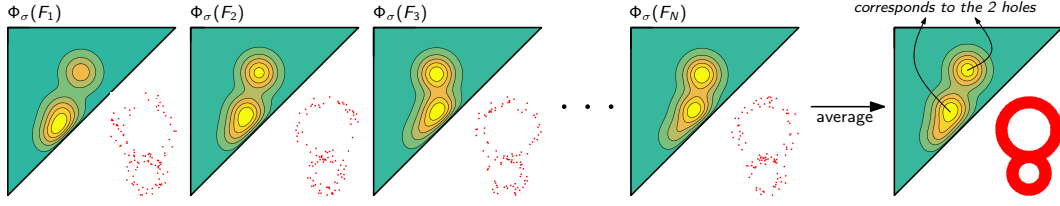

**Fig. 2:** Visualization of the mean PSS function (right) taken over 30 samples from a double-annulus (cf. [19]).

$f : \mathcal{R} \to \mathcal{S}$ that maps $(p_1, \ldots, p_N) \in \mathcal{R}$ to the persistence diagram $\{p_i : 1 \leq i \leq N$ if $p_i \notin \partial\Omega\} \in \mathcal{S}$. We note that for all $D = \{p_1, \ldots, p_n\} \in \mathcal{S}$, with $n \leq N$, there exists an $X \in \mathcal{R}$, e.g., $X = (p_1, \ldots, p_n, 0, \ldots, 0)$, such that $f(X) = D$; this implies $\mathcal{S} = f(\mathcal{R})$. Next, we show that $f$ is 1-Lipschitz continuous w.r.t. the 1-Wasserstein distance on persistence diagrams, i.e.,

$$\forall X = (p_1, \ldots, p_N), Y = (q_1, \ldots, q_N) \in \mathcal{R} : d_{W,1}(f(X), f(Y)) \leq d(X, Y),$$

where we defined $d$ as $\inf_\gamma \sum_{1 \leq i \leq N} \|p_i - \gamma(p_i)\|_\infty$ with $\gamma$ ranging over all bijections between $\{p_1, \ldots, p_N\}$ and $\{q_1, \ldots, q_N\}$. In other words, $d$ corresponds to the 1-Wasserstein distance *without* allowing matches to the diagonal. Now, by definition, $d_{W,1}(f(X), f(Y)) \leq d(X, Y)$, because all bijections considered by $d$ are also admissible for $d_{W,1}$. Since thus $\mathcal{R}$ is compact and $f$ is continuous, we have that $\mathcal{S} = f(\mathcal{R})$ is compact as well. $\square$

We refer to the kernel of Eq. (5) as the *universal persistence scale-space (u-PSS)* kernel.

**Remark.** While we prove Prop. 1 for the PSS kernel in Eq. (3), it obviously also holds for $k_\sigma^U$, since exponentiation does neither invalidate measurability nor boundedness.

**Relation to persistence landscapes.** As the feature map $\Phi_\sigma$ of Eq. (2) defines a function-valued summary of persistent homology in the Hilbert space $L^2(\Omega)$, the results on probability in Banach spaces [14], used in [4] for persistence landscapes, naturally apply to $\Phi_\sigma$ as well. This includes, for instance, the law of large numbers or the central limit theorem [4, Theorems 9,10]. Conversely, considering a persistence landscape $\lambda(D)$ as a function in $L^2(\mathbb{N} \times \mathbb{R})$ or $L^2(\mathbb{R}^2)$ yields a positive definite kernel $\langle \lambda(\cdot), \lambda(\cdot) \rangle_{L^2}$ on persistence diagrams. However, it is unclear whether a universal kernel can be constructed from persistence landscapes in a way similar to the definition of $k_\sigma^U$. In particular, we are not aware of a proof that the construction of persistence landscapes, considered as functions in $L^2$, is continuous with respect to $d_{W,q}$ for some $1 \leq q \leq \infty$. For a more detailed treatment of the differences between $\Phi_\sigma$ and persistence landscapes, we refer the reader to [20].

## 4 Experiments

We first describe a set of experiments on synthetic data appearing in previous work to illustrate the use of the PSS feature map $\Phi_\sigma$ and the universal persistence scale-space kernel on two different tasks. We then present two applications on real-world data, where we assess differences in the persistent homology of functions on 3D surfaces of lateral ventricles and corpora callosa with respect to different group assignments (i.e., age, demented/non-demented). In all experiments, filtrations and the persistence diagrams are obtained using DIPHA[2], which can directly handle our types of input data. Source code to reproduce the experiments is available at https://goo.gl/KouBPT.

### 4.1 Synthetic data

**Computation of the mean PSS function.** We repeat the experiment from [19, 4] of sampling from the union of two overlapping annuli. In particular, we repeatedly ($N = 30$ times) draw samples of size 100 (out of 10000), and then compute persistence diagrams $F_1, \ldots, F_N$ for 1-dim. features by considering sublevel sets of the distance function from the points. Finally, we compute the mean of the PSS functions $\Phi_\sigma(F_i)$ defined by the feature map from Eq. (2). This simply amounts to computing $1/N \cdot \Phi_\sigma(F_1 \cup \cdots \cup F_N)$. A visualization of the pointwise average, for a fixed choice of $\sigma$, is shown in Fig. 2. We remind the reader that the convergence results used in [4] equally hold for this feature map, as explained in Section 3. In particular, the above process of taking means converges to the expected value of the PSS function. As can be seen in Fig. 2, the two 1-dim. holes manifest themselves as two "bumps" at different positions in the mean PSS function.

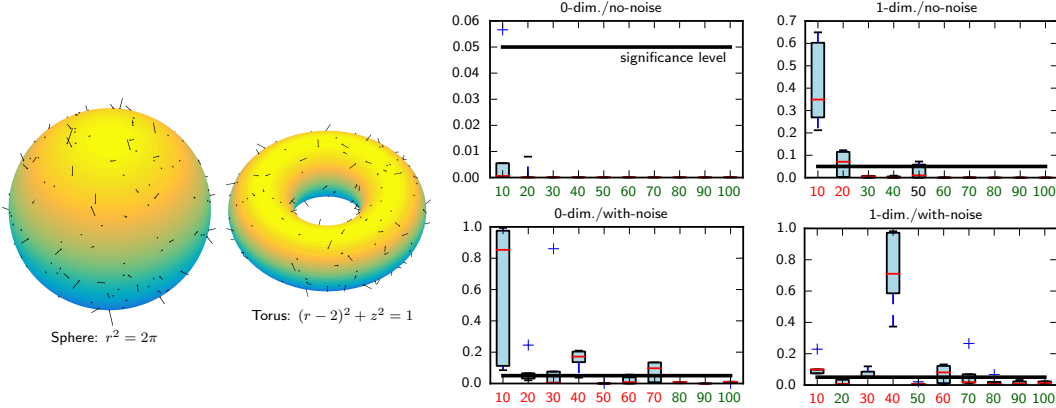

**Fig. 3:** *Left*: Illustration of one random sample (of size 200) on a sphere and a torus in $\mathbb{R}^3$ with equal surface area. To generate a noisy sample, we add Gaussian noise $\mathcal{N}(0, 0.1)$ to each point in a sample (indicated by the vectors). *Right*: Two-sample hypothesis testing results ($H_0 : P = Q$ vs. $H_A : P \neq Q$) for 0- and 1-dim. features. The box plots show the variation in *p*-values (*y*-axis) over a selection of values for $\sigma$ as a function of increasing sample size (*x*-axis). Sample sizes for which the median *p*-value is less than the chosen significance level (here: 0.05) are marked green, and red otherwise.

**Torus vs. sphere.** In this slightly more involved example, we repeat an experiment from [4, Section 4.3] on the problem of discriminating between a sphere and a torus in $\mathbb{R}^3$, based on random samples drawn from both objects. In particular, we repeatedly ($N$ times) draw samples from the torus and the sphere (corresponding to measures $P$ and $Q$) and then compute persistence diagrams. Eventually, we test the null-hypothesis $H_0 : P = Q$, i.e., that samples were drawn from the same object; cf. [4] for a thorough description of the full setup. We remark that our setup uses the Delaunay triangulation of the point samples instead of the Coxeter–Freudenthal–Kuhn triangulation of a regular grid as in [4].

Conceptually, the important difference is in the two-sample testing strategy. In [4], two factors influence the test: (1) the choice of a functional to map the persistence landscape to a scalar and (2) the choice of test statistic. Bubenik chooses a *z*-test to test for equality between the mean persistence landscapes. In contrast, we can test for true *equality in distribution*. This is possible since universality of the kernel ensures that the MMD of Eq. (1) is a metric for the space of probability measures on persistence diagrams. All *p*-values are obtained by bootstrapping the test statistic under $H_0$ over $10^4$ random permutations. We further vary the number of samples/object used to compute the MMD statistic from $N = 10$ to $N = 100$ and add Gaussian noise $\mathcal{N}(0, 0.1)$ in one experiment. Results are shown in Fig. 3 over a selection of u-PSS scales $\sigma \in \{100, 10, 1, 0.1, 0.01, 0.001\}$. For 0-dimension features and no noise, we can always reject $H_0$ at $\alpha = 0.05$ significance. For 1-dim. features and no noise, we need at least 60 samples to reliably reject $H_0$ at the same level of $\alpha$.

## 4.2 Real-world data

We use two real-world datasets in our experiments: (1) 3D surfaces of the corpus callosum and (2) 3D surfaces of the lateral ventricles from neotates. The corpus callosum surfaces were obtained from the longitudinal dataset of the OASIS brain database[3]. We use all subject data from the *first* visit, and the grouping criteria is disease state: dementia vs. non-dementia. Note that the demented group is comprised of individuals with very mild to mild AD. This discrimination is based on the clinical dementia rating (CDR) score; Marcus et al. [17] explain this dataset in detail. The lateral ventricle dataset is an extended version of [3]. It contains data from 43 neonates. All subjects were repeatedly imaged approximately every 3 months (starting from 2 weeks) in the first year and every 6 months in the second year. According to Bompard et al. [3], the ventricle growth is the dominant effect and occurs in a non-uniform manner most significantly during the first 6 months. This raises the question whether age also has an impact on the shape of these brain structures that can be detected by persistent homology of the HKS (see *Setup* below, or Section 2) function. Hence, we set our grouping criteria to be developmental age: $\leq$ 6 months vs. > 6 months. It is important to note that the heat kernel signature is *not* scale-invariant. For that reason, we normalize the (mean-subtracted) configuration matrices (containing the vertex coordinates of each mesh) by their Euclidean norm, cf. [9]. This ensures that our analysis is not biased by growth (scaling) effects.

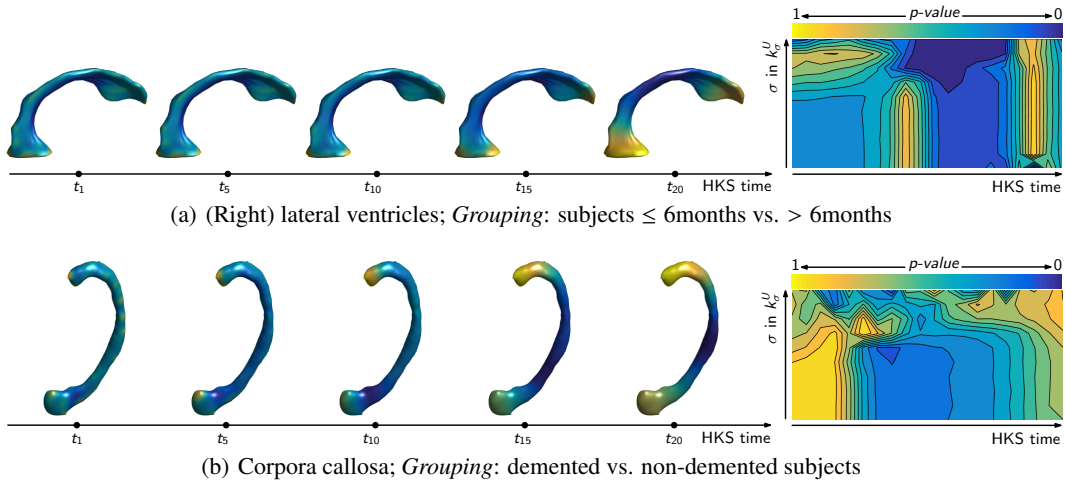

(a) (Right) lateral ventricles; *Grouping*: subjects ≤ 6months vs. > 6months

(b) Corpora callosa; *Grouping*: demented vs. non-demented subjects

**Fig. 4:** *Left:* Effect of increasing HKS time $t_i$, illustrated on one exemplary surface mesh of both datasets. *Right:* Contour plots of *p*-values estimated via random permutations, shown as a function of the u-PSS kernel scale $\sigma$ and the HKS time.

**Setup.** We follow an experimental setup, similar to [16] and [20], and compute the heat kernel signature [27] for various times $t_i$ as a function defined on the 3D surface meshes. In all experiments, we use the proposed kernel u-PSS kernel $k_\sigma^U$ of Eq. (5) and vary the HKS time $t_i$ in $1 = t_1 < t_2 < \cdots < t_{20} = 10.5$; Regarding the u-PSS kernel scale $\sigma_i$, we sweep from $10^{-9} = \sigma_1 < \cdots < \sigma_{10} = 10^1$. Null- ($H_0$) and alternative ($H_A$) hypotheses are defined as in Section 2 with two samples of persistence diagrams $\{F_i\}_{i=1}^m, \{G_i\}_{i=1}^n$. The test statistic under $H_0$ is bootstrapped using $B = 5 \cdot 10^4$ random permutations. This is also the setup recommended in [13] for low samples sizes.

**Results.** Figure 4 shows the estimated *p*-values for both datasets as a function of the u-PSS kernel scale and the HKS time for 1-dim. features. The false discovery rate is controlled by the Benjamini-Hochberg procedure. On the lateral ventricle data, we observe *p*-values < 0.01 (for the right ventricles), especially around HKS times $t_{10}$ to $t_{15}$, cf. Fig. 4(a). Since the results for left and right lateral ventricles are similar, only the *p*-values plots for the right lateral ventricle are shown. In general, the results indicate that, at specific settings of $t_i$, the HKS function captures salient shape features of the surface, which lead to statistically significant differences in the persistent homology. We do, however, point out that there is no clear guideline on how to choose the HKS time. In fact, setting $t_i$ too low might emphasize noise, while setting $t_i$ too high tends to smooth-out details, as can be seen in the illustration of the HKS time on the left-hand side of Fig. 4. On the corpus callosum data, cf. Fig. 4(b), no significant differences in the persistent homology of the two groups (again for 1-dim. features) can be identified with *p*-values ranging from 0.1 to 0.9. This does not allow to reject $H_0$ at any reasonable level.

## 5 Discussion

With the introduction of a universal kernel for persistence diagrams in Section 3, we enable the use of this topological summary representation in the framework of embedding probability measures into reproducing kernel Hilbert spaces. While our experiments are mainly limited to two-sample hypothesis testing, our kernel allows to use a wide variety of statistical techniques and learning methods which are situated in that framework. It is important to note that our construction, via Theorem 1, essentially depends on a restriction of the set $\mathcal{D}$ to a compact metric space. We remark that similar conditions are required in [4] in order to enable statistical computations, e.g., constraining the support of the persistence landscapes. However, it will be interesting to investigate which properties of the kernel remain valid when lifting these restrictions. From an application point of view, we have shown that we can test for a statistical difference in the distribution of persistence diagrams. This is in contrast to previous work, where hypothesis testing is typically limited to test for specific properties of the distributions, such as equality in mean.

**Acknowledgements.** This work has been partially supported by the Austrian Science Fund, project no. KLI 00012. We also thank the anonymous reviewers for their valuable comments/suggestions.

## Footnotes

[1] The $q$-Wasserstein metric is defined as $d_{W,q}(F, G) = \inf_\gamma (\sum_{x \in F} \|x - \gamma(x)\|_\infty^q)^{1/q}$, where $\gamma$ ranges over all bijections from $F \cup D$ to $G \cup D$, with $D$ denoting the multiset of diagonal points $(t, t)$, each with countably infinite multiplicity.

[2]available online: https://code.google.com/p/dipha/

[3]available online: http://www.oasis-brains.org

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
