[Reviews · NeurIPS 2015]

Submitted by Assigned_Reviewer_1

Summary:

topological data analysis is a field which studies the geometric "shape" of a data set, using methods from algebraic topology. A tool that has been proven useful is the so-called persistent homology together the persistence diagrams. One is particularly interested in computing these persistence diagrams as they give the "topological summary" of the data shape. However a statistical treatment of these persistent diagrams is rather computationally expensive and difficult. A way to circumnavigate this difficulty goes by embedding probability measure on the space of persistence diagrams into a a reproducing kernel Hilbert space, since Hilbert spaces are highly regular and have an abundance of available well-studied methods that largely simplify calculations. Such an embedding can be achieved if the corresponding kernel is "universal", via the so-called "mean map". The paper proves that a slight modification of the recently introduced "PSS kernel", called the "u-PSS kernel", has the property of being "universal" (with respect to the 1-Wasserstein metric) [prop 2]. For example, one can then test whether two different samples exhibit significantly different homological features, while avoiding the direct computation of persistence diagrams and reducing it to the evaluation of the corresponding kernel.

Major negative points

- The paper is generally hard to read for people not so familiar with algebraic topology or functional analysis. Despite the background chapter, several rather more advanced notions or results in functional analysis are assumed familiar and not explained in further detail. - Many relevant definitions, argumentations from other papers and calculations are rather sketchy or even omitted.

- The core argumentation of the paper is not written down in a concise chronological "start-to-finish argument", rather the different pieces of the argument are spread out throughout the paper, randomly mixed in with different information of secondary importance which makes the reading at times rather confusing.

- The actual contribution is the proof of proposition 2, which in turn just boils down to proving that (\mathcal{S},d_{W,1}) is a compact metric, due to Theorem 1. In my opinion, the actual contribution is not that significant. The author mainly put together several already-published definitions and results, to achieve his result without introducing anything novel.

Positive points

- the field of topological data analysis is a rather new and exciting area with many interesting applications, and the result seems rather exciting. As algebraic topology is a rather powerful tool, however not frequently used due to its computational complexity, a result leading to implementations of reduced complexity can lead to interesting and very diverse implications. - the experiments /implementations illustrate well the generality of theoretical result, and the diversity of its applications.

Final conclusion: 5/10 (weak rejection)
Summary: Not enough theoretical and empirical content, writing not so great.

Submitted by Assigned_Reviewer_2

Summary:

The authors investigate the theoretical foundation of applying

Reproducing Kernel Hilbert Space based technique in expressing the topological invariances of the persistent homology. That investigation focuses on the universality of the potential kernel representation. They demonstrate the usefulness of their approach for example in two-sample hypothesis testing.

General remarks: --- The statement of Proposition 2 can be reinterpreted since the exponential mapping applied there can be taken as a projection from the manifolds of the positive semidefinite kernels onto the corresponding tangent plane. Thus investigating the the properties of that manifold might yield some further ways to other kind of generalization. For example looking at the sensitivity of universality with respect to the local properties of that manifold might allows to extend the scope of that Proposition.

--- In the paper only the solution of the heat diffusion problem is considered as a source of a possible kernel representation. What about other possibilities, for example some results in discrete differential geometry may provide other feature mapping exploitable in this context.

--- The proposed approach is demonstrated via discriminative hypotheses testing. The usefulness of those tests can be checked against discriminative learning methods which can provide confirmation to the applicability. I think that kind of control tests should be carried out for example on a sample of randomly chosen point clouds representing balls and tori.

Summary: Pros: The authors provide a comprehensive introduction of the topological data analysis and the potential kernel representation of the corresponding problems.

Cons:

The authors do not give a clear view why the universality of a kernel can really help in a learning problem. Showing up a family of problems where this approach has clear advantages can make the paper significantly stronger.

Submitted by Assigned_Reviewer_3

SUMMARY

This paper proves universality of a variant of a recently published kernel on persistence diagrams, which are frequently used to encode the persistent homology of data. Universality of the kernel enables the use of kernelized hypothesis testing, which is applied to studying the topology of the underlying data space of both synthetic and real data.

The paper is very well written, although I would wish for it to be more self contained. The paper is sound and interesting, and opens up to kernel methods for topological data analysis which is novel as far as I know, except for the recently published kernel. The previous paper reduces the novelty, and the experimental validation does not shine light on whether the paper actually improves on the state-of-the art as there is no comparison to other methods. On the other hand, I think the paper might generate interesting discussions in the NIPS community.

DETAILS

*Completeness*

It would have been nice to see a more detailed discussion of topological data analysis, and persistence diagrams + Wasserstein distances in particular, as these are not mainstream topics at NIPS. I urge the authors to include this in their final version. Space could e.g. be taken from the current discussion of related work which is very much focused on the computation of means of persistence diagrams (no need to leave out references, just shorten the description, which is not crucial in this paper).

Similarly, it would be very helpful if you include definitions of terms like "birth-death bounded" (although I appreciate the reference). The average NIPS attendee might like to read your paper, but likely would not look up terms like this, so including definitions will largely increase your audience.

*Proposition 2*

I have a question regarding the proof of Proposition 2. It is not clear to me why f(\mathcal{R}) = \mathcal{S}, but this probably has to do with the lack of including the definition of "birth-death bounded"?

*Experiments*

It would have been nice to see a comparison with other methods, e.g. permutation tests comparing means or variances of the data sets with respect to the Wasserstein distance, or comparing to shape analysis of the surfaces in section 4.2. It is left unclear whether the method improves on the state-of-the-art.

TYPOS

line 094: form -> from on page 3 your reference to [13] should probably come the minute you start talking about MMD, as this is not a contribution of your paper line 222: an function -> a function
Summary: Interesting and well written paper which proves universality of a variant of a previously published kernel for topological data analysis. Novelty is a bit limited and experimental validation lacks comparison with other methods.

Submitted by Assigned_Reviewer_4

(This is a "light" review, so no detailed comments, but a few remarks.)

* "total multiplicities are bounded by N" in the second paragraph of Section 3

confused me. I thought you meant the maximum multiplicity of any point is at

most N, but it seems (after the discussion of the bounded subset of (R^2)^N

that you are working with) that you mean the sum of multiplicities of

off-diagonal points is at most N. It would be good to clarify this upfront.

* In the paper, you down-play the requirements on bounded persistence and, more

importantly, on bounded multiplicities. But, of course, it's a serious

limitation. Especially odd is the argument that because we are interested in

finite processes, we don't have to care. The type of theorem you present

would probably be most useful in relating the persistence diagram of some

smooth function (possibly perturbed by noise) to what you get from a finite

approximation (to get inference-type results). But the smooth perturbed

function is unlikely to satisfy the bounded multiplicities requirements. It

would probably be helpful to add a discussion of this in the paper to alert a

non-expert reader of potential issues.

* Figure 2 is more confusing than necessary because of its 3D view of the

function on the plane.

The functions should be symmetric, but it's not clear

from the figure that they are. Maybe it would be better to show them as

2-dimensional colormaps?

* I'm badly confused by all the talk of 0-dimensional features recovering

the two holes, in Section 4.1. How is this possible? You seem to be taking

sublevel sets of the distance functions, and you don't seem to be working

with "extended persistence." Do 0-dimensional features mean something other

than "we are looking at 0-dimensional homology"? There are more issues like

this in Section 4. This really needs to be clarified: it's very confusing.
Summary: The paper is an important contribution to bringing statistical foundations to the study of persistent homology (an algebraic topological descriptor of scalar functions). Roughly, it applies kernel methods and maps a persistence diagram to a scalar function over a plane. It shows that the resulting kernel is universal and that this fact is useful in practice. The results are strong and important, so I recommend the paper for publication.

Author Feedback
Author rebuttal: We thank all reviewers for their reviews and valuable comments. Below we respond to concerns/questions.

==Reviewer 1==

We have adjusted (to the extent possible) Sect. 2 (Background) to be more precise with respect to our description of (1) filtrations and the (2) embedding of probability measures into reproducing kernel Hilbert spaces. This section should now be more self-contained and easier to follow.

We also highlight that our main contribution does *not* boil down to the proof of Prop. 2. Conceptually, our contribution is to enable a statistical treatment of persistence diagrams via kernel methods. This is, to the best of our knowledge, a novel idea. Technically, our contribution is the insight that the original PSS kernel of Reininghaus et al. [7], constructed via an explicit feature map, allows us to leverage Christman et al.'s [7] results to demonstrate universality. Along this way, Prop. 1 and Prop. 2 are only a means to an end to establish the prerequisites of Theorem 1. We believe this result is also not trivial, since Prop. 1/2 depend on a careful analysis of (1) the PSS feature map from [7] and (2) restrictions on the diagrams (e.g., bounded multiplicity) to achieve compactness of the space. In case of persistence landscapes for instance, Prop. 1 is invalidated since the construction of landscapes (as L^2 functions) is not continuous w.r.t. any d_{W,q}.

We have adjusted the "Contribution" part of Sect. 1 to better highlight the relevant parts. Also, a concise "start-to-finish" argument, as requested by the reviewer, can
(and will be) included in Sect. 1, linking to the corresponding sections of the paper.

==Reviewer 2==

It is indeed true that heat diffusion, as a starting point to develop an alternative representation for persistence diagrams, is only one possible way. Our focus, however, is on the subsequent step of allowing statistical computations with this "new" representation in a kernel-based setting. We do agree that both topics are intimately related, and different mappings would clearly be possible. For instance, as mentioned in our response to Rev. #1, while persistence landscapes (as L^2 functions) are not continuous w.r.t. any d_{W,q}, the variant persistence silhouettes might allow for results similar to ours.

Regarding "control tests" to validate applicability of the kernel: we will include such an experiment (with SVMs) in the final version and report cross.-val results.

==Reviewer 3==

We agree that a definition of "birth-death boundedness" enhances presentation of the material and makes it more accessible to the reader. This is now included; we have also shortened "Related work" in favor of a more detailed description of persistent homology.

Furthermore, Sect. 3 now better describes the relation between \mathcal{D}_q (i.e., the space of persistence diagrams), R and N, since they are closely related.

Regarding Prop. 2: To obtain \mathcal{S} = f(\mathcal{R}), note that, by definition, f(\mathcal{R}) \subset \mathcal{S}. Now, for all D \in \mathcal{S} there exists a R \in \mathcal{R} such that D = f(R). Consequently, we get \mathcal{S} = f(\mathcal{R}).

We'd like to thank the reviewer for pointing out the minor mistakes; they are now fixed.

==Reviewer 4==

The description of N is indeed misleading due to our wording. We have adjusted/clarified this. As correctly presumed by the reviewer, N denotes the total sum of multiplicities of the points in a diagram; this is typically referred to as the "total multiplicity".

Regarding the boundedness of N: boundedness results from a discretization of the input. For (1) meshes, N is bounded by the #simplices (i.e., vertices, edges, triangles); for (2) images, N is bounded by the #pixel; for (3) point clouds, N is bounded by the #simplices in the complex (e.g., Vietoris-Rips).

Regarding boundedness of R: for images, R is bounded by the value range of the pixel; for point clouds, R is bounded by the diameter of the cloud(s). In the case of meshes, however, where we work with functions on the simplicial complex, the situation is indeed more challenging. Bounding R might have to be enforced artificially, e.g., by considering the heat-kernel signature (HKS) only after a certain amount of time to avoid pathological cases. We will clarify this in a final version, to make the reader aware of this limitation.

In Fig. 2, the reviewer correctly observes that 0-dimensional diagrams do NOT allow us to identify holes. This was a typo in the text; in fact, the 1-dimensional diagrams are shown (not 0-dimensional). We have fixed this. Also, we appreciate the comment on the visualization of the PSS mapping and we will show heat-maps instead.

==Reviewer 6==

We have added, at the location of the URL where our source code resides, a more thorough introduction to the background material and specific references to papers where those concepts are introduced in an way that is accessible to the non-expert reader.